# Particle Swarm Optimization and Real-Road/ Driving-Cycle Analysis Based Powertrain System Design for Dual Motor Coupling Electric Vehicle

**Chao Ma [1],\*** ⓘ**, Shiwei Jin [1], Kun Yang [1,2]** ⓘ**, Di Tan [1]** ⓘ**, Jie Gao [1] and Dechao Yan [1]**

[1] College of Transportation and Vehicle Engineering, Shandong University of Technology, Zibo 255000, China; 18502100129@stumail.sdut.edu.cn (S.J.); yangkun_sdut@163.com (K.Y.); tandi@sdut.edu.cn (D.T.); gaojie18853584120@163.com (J.G.); yanchao19970311@163.com (D.Y.)

[2] Shandong Yiwei Automobile Technology Co., Ltd., Zibo 255000, China

\* Correspondence: mc@sdut.edu.cn; Tel.: +86-152-6438-2636

**Abstract:** In this study, a planetary gear based dual motor coupling electric vehicle is proposed, which achieves higher system efficiency by enabling motor working under high operating efficiency area. Firstly, the dynamic characteristics of the proposed configuration are analyzed and the reasonable working modes are established. Secondly, the optimal dual motor parameters are derived according to the statistical analysis on the typical driving conditions and the collected real road driving data. Especially, the optimal parameters of planetary gear and final transmission ratio are obtained using particle swarm optimization algorithm. Finally, based on the developed mode shift algorithm, the dual motor coupling full vehicle model is developed and the vehicle performance is analyzed using MATLAB/Simulink. For the UDDS (Urban Dynamometer Driving Schedule) driving cycle, it is seen from the simulation results of motor operating points that the proposed dual motor configuration is mostly operated under the high efficiency range, and the power consumption is significantly reduced by 7.6% compared with the single motor configuration. For the NEDC (New European Driving Cycle), WLTC (Worldwide Harmonized Light Vehicles Test Cycle) and real road driving conditions, the proposed dual motor configuration also achieves system efficiency improvement of 5.0~16.3%, which confirms the validity of the proposed configuration and its corresponding parameter matching and control algorithm development.

**Keywords:** dual motor configuration; planetary gear; driving condition analysis; real road driving data collection; parameter matching; particle swarm optimization

## 1. Introduction

With the global energy shortage and the increasingly serious environmental pollution, the popularization and promotion of new energy vehicles become increasingly important [1,2]. As one of the new energy vehicles, electric vehicle (EV) has high energy efficiency, which can truly achieve pollution-free and zero emission. In addition, electric energy can be widely and conveniently obtained. So the promotion and development of EV can effectively reduce the above problems [3].

Since the battery technology has not yet broken through the bottleneck, the development of EV is confined by the short driving range and relatively long charging time. Therefore, the reasonable configuration and parameter matching has become extremely important at the design stage of EV. In the early stage, the pure electric vehicles in the market were mostly single motor and fixed gear ratio configuration. This configuration is widely used because of its simple configuration and easy to control. For this configuration, due to the narrow motor high efficiency range, the EV can not obtain high system

efficiency at low speed or high speed driving conditions. Therefore, some researches are investigated on the improvement of EV configuration and system efficiency. For example, the single motor single gear configuration is replaced by adding a two speed transmission to the EV, and reasonable shift strategy is formulated to obtain high system efficiency. The application of two speed transmission improves the motor high efficiency operating range through the optimal shift control and proper parameter matching of the target system [4–7].

Recently, the research on the multi motor configuration and reasonable energy management strategy to improve the vehicle economy has become a trend. Compared with single motor configuration, the multi motor configuration is proposed, which can solve the problem of power redundancy and realize the high efficiency operation of motor. Xiong, et al. [8] and Mutoh, et al. [9] adopted the front and rear axle arrangement of motor to improve the braking and driving performance of the vehicle. Shi, et al. [10] used two motors with different power in front and rear axles to deal with different driving conditions, and a genetic algorithm is applied to determine the optimal torque distribution and improve the economy. Meanwhile, the introduction of hub motor also improves the economy by using reasonable control strategies [11–14]. In recent years, an increasing number of researches have been carried out to reduce energy consumption by adopting multi motor coupling configuration. Zhu, et al. [15] proposed a dual motor and two gear coupling configuration of EV, which achieves the motor optimal working and vehicle performance improvement by working under single motor working mode and dual motor torque coupling mode. Zhang, et al. [16] put forward an improved rule-based control strategy for a dual motor coupling configuration. Hu, et al. [17] used planetary gear as the dynamic coupling mechanism, which investigated speed coupling and torque coupling to improve the dynamic and economic performance. Consequently, reasonable configuration design, accurate system parameter matching and optimization, energy management strategy should be investigated to achieve better vehicle energy consumption reduction of EV.

In this paper, a planetary gear coupling mechanism based dual motor configuration is proposed, which can obtain high system efficiency improvement through optimized system parameters matching and energy management strategy. First, by the statistical analysis of typical driving conditions and real road driving conditions, the accurate optimal parameter matching of the two motors are carried out with the consideration of two motors working under mostly operated high efficiency range for different driving conditions. Then, a particle swarm optimization algorithm is used to optimize the transmission system parameters, which further reduces the energy cost and increases the driving range effectively. In order to verify the proposed system matching and optimization, the powertrain system model is developed using MATLAB/Simulink. Especially, the statistical analysis based mode shift control and optimal dual motor speed operating control are developed. Finally, the correctness of configuration, matching results and energy management strategy are verified through comparative analysis.

## 2. Dynamic Characteristic Analysis for the Planetary Gear Based Dual Motor Coupling EV

Since the single motor electric vehicle has the problem of low efficiency, the design of electric vehicle with multi motors and multi gears has become an important trend in the near future for its effectively improvement of system efficiency. In this study, the planetary gear based dual motor coupling mechanism is introduced, which can improve the system efficiency, reduce the power waste and energy consumption.

### 2.1. Parameters of the Target Vehicle

The dual motor coupling configuration electric vehicle is designed on the basis of single motor and single gearbox configuration. The vehicle parameters and performance indicators are shown in Table 1.

**Table 1.** Target vehicle parameters and performance indicators.

| Item | Value | Unit |
|---|---|---|
| Full load mass, $m$ | 1750 | kg |
| Windward area, $A$ | 2.35 | m$^2$ |
| Air drag coefficient, $C_D$ | 0.3 | - |
| Rolling resistance coefficient, $f$ | 0.012 | - |
| Tire rolling radius, $r$ | 0.31 | m |
| Transmission efficiency, $\eta_t$ | 0.95 | - |
| Correction coefficient of rotating mass, $\delta$ | 1.04 | - |
| Final reduction gear ratio, $i_0$ | 8.28 | - |
| Rated/maximum speed | 3300/10,000 | rpm |
| Rated/maximum torque | 116/230 | Nm |
| Rated/peak power | 40/80 | kw |
| Maximum vehicle speed | 130 | km/h |
| Acceleration time (0~50 km/h/0~100 km/h) | 4.5/12 | s |
| Maximum climbing grade (20 km/h) | 28% | - |
| Driving range of constant speed cruise (60 km/h) | 480 | km |

## 2.2. Dynamic Analysis of Planetary Gear Coupling Mechanism

The configuration proposed in this paper is shown in Figure 1. The driving modes can be divided into two types: single motor driving mode and dual motor driving mode. In the single motor driving mode, the two motors are designed suitable for low speed and medium speed driving conditions respectively; in the high speed condition, the two motors are cooperated and working under dual motor speed coupling driving mode, which realizes the motor operation under high efficiency range and improves the full vehicle economy. In this configuration, the motor-generator 1 (MG1) is connected with the sun gear of the planetary gear, and the motor-generator 2 (MG2) is connected with the ring gear through a fixed gear. The working status of each component under the three working modes is shown in Table 2.

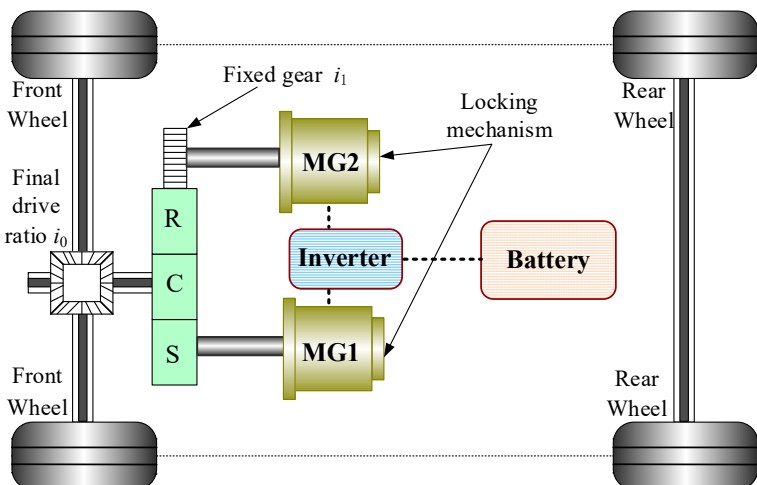

**Figure 1.** The configuration of dual-motor coupling.

**Table 2.** Component working status under different working modes.

| Working Modes | MG1 | MG2 | Sun Gear | Ring Gear |
|---|---|---|---|---|
| MG1 driving | ON | OFF | ON | Locked |
| MG2 driving | OFF | ON | Locked | ON |
| Dual motor speed coupling driving | ON | ON | ON | ON |

According to the theory of leverage analysis, the dynamic characteristics of each working mode are derived as follows:

(1) MG1 driving: low speed (urban) driving conditions. In this working mode, the ring gear is locked, and the power is input by the sun ring and output by the planetary carrier, which acts as a reduction gear. The speed and torque relationship between the planetary carrier and the sun gear are as follows.

$$\omega_c = \frac{\omega_{MG1}}{1+k} \tag{1}$$

$$T_c = (1+k)T_{MG1} \tag{2}$$

(2) MG2 driving: medium speed (suburban) driving conditions. The sun gear is locked, and the power is input by the ring gear and output by the planetary carrier, which also acts as a reduction gear. The speed and torque relationship between the planetary carrier and the ring gear are as follows.

$$\omega_c = \frac{k \cdot \omega_{MG2}}{i_1 \cdot (1+k)} \tag{3}$$

$$T_c = \frac{i_1 \cdot (1+k)}{k} T_{MG2} \tag{4}$$

(3) Dual-motor speed coupling driving: high speed driving conditions. In this mode, the power is input by the sun gear and ring gear, and output by the planetary carrier after coupling with the planetary gear. The dynamic relationship is as follows.

$$(1+k)\omega_c = \omega_{MG1} + \frac{k}{i_1} \cdot \omega_{MG2} \tag{5}$$

$$T_c = (1+k)T_{MG1} = \frac{i_1 \cdot (1+k)}{k} T_{MG2} \tag{6}$$

where $T_c$ is the torque of planetary carrier, Nm; $\omega_c$ is the rotational speed of planetary carrier, rad/s; $\omega_{MG1}$ is rotational the speed of MG1, rad/s; $T_{MG1}$ is the torque of MG1, Nm; $\omega_{MG2}$ is rotational the speed of MG2, rad/s; $T_{MG2}$ is the torque of MG2, Nm; $k$ is the characteristic parameter of the planetary gear (Ring/Sun); $i_1$ is the fixed gear ratio.

From the working characteristics of planetary gear mechanism, the following analysis can be concluded. For the MG1 driving mode, the ring gear is locked and the power is transmitted from the sun gear to the carrier (output), which has the maximum transmission ratio and maximum output torque. Therefore, the MG1 driving mode is suitable for the low speed and large torque driving condition. For the MG2 driving mode, the transmission ratio is smaller compared with the MG1 driving mode, which meets the medium speed condition. For the dual motor coupling driving mode, the MG1 and MG2 are cooperate together with the optimal motor speed coupling, which has higher system efficiency at the high speed driving condition.

## 3. Parameter Matching and Optimization of Powertrain System

First, according to the dynamic and economic performance, the primary motor and battery parameters are calculated. Second, the optimal parameters of the two motors are obtained through the statistics analysis on the real road and multi driving conditions. Finally, the transmission parameters are derived using particle swarm optimization algorithm. Based on the above optimal work, the optimal parameters of the planetary gear based dual motor coupling EV are obtained.

*3.1. Primary Parameter Matching of the Target Vehicle*

At this part, the primary motor and battery parameters are calculated.

### 3.1.1. Vehicle Demand Power

The vehicle demand motor is calculated according to the vehicle maximum speed, acceleration performance, and maximum climbing gradient [18,19]:

$$P_{max1} \geq \frac{1}{\eta_t}\left(\frac{mgf}{3600}u_{amax} + \frac{C_DA}{76140}u_{amax}{}^3\right) \tag{7}$$

$$P_{max2} \geq \frac{1}{\eta_t}\left(\frac{\delta mu_m^2}{3600 \times 3.6dt}\left[1 - \left(\frac{t_m - dt}{t_m}\right)^x\right] + \frac{mgfu_m}{3600} + \frac{C_DAu_m{}^3}{76140}\right) \tag{8}$$

$$P_{max3} \geq \frac{u_i}{3600\eta_t}\left(mgfcos\alpha_{max} + mgsin\alpha_{max} + \frac{C_DAu_i{}^2}{21.15}\right) \tag{9}$$

where $P_{max1}$ is the power required to meet the maximum speed; $P_{max2}$ is the power required to meet the acceleration time; $P_{max3}$ is the power required to meet the maximum climbing gradient; $u_{amax}$ is the maximum vehicle speed; $u_i$ is the constant speed of the climbing process; $x$ is the fitting coefficient, generally is 0.5; $t_m$ is the acceleration time, $u_m$ is the final speed after acceleration.

Therefore, the total demand power of the vehicle is:

$$P_{max} \geq max\{P_{max1}, P_{max2}, P_{max3}\} \tag{10}$$

The total demand power of the vehicle after rounding is 80 kw.

### 3.1.2. Power Battery

At present, the rated voltage of lithium-ion batteries commonly used in the market is generally 3.7 V. Therefore, the rated voltage of the battery pack is designed to $U = 3.7$ V $\times$ 98 $\approx$ 360 V. The battery capacity is calculated based on the driving range of the target vehicle. From Table 1, it is seen that the vehicle should have a driving range of 480 km under constant speed driving of 60 km/h. The battery required energy is as follows:

$$W_1 = P_1 \times t = \frac{u_a}{3600\eta_t}\left(mgf + \frac{C_DA}{21.15}u_a{}^2\right) \times \frac{S}{u_a} \tag{11}$$

where $P_1$ is the power required for driving at a constant speed of 60 km/h; $S$ is the driving range; $W_1$ is the energy required.

The actual discharged energy of the battery pack is:

$$W_{act} = \frac{U \times C \times \zeta_{SOC}}{1000} \geq W_1 \tag{12}$$

where $U$ is battery voltage; $\zeta_{SOC}$ is the effective capacity discharge coefficient of battery, generally is 0.9; $C$ is the battery capacity.

According to the calculation, the battery capacity $C$ is 141 Ah. Considering the need for some redundancy, the battery capacity is 145 Ah.

### 3.2. Real-Road/Driving-Cycle Analysis Based Parameter Matching of Dual Motor

In order to achieve the optimal high system efficiency of the dual motor configuration, the parameters of the two motors are derived through the motor high efficiency operating range under corresponding driving conditions. Based on the motor optimal operating, the vehicle can take full advantage of the flexible and variable modes for the dual motor configuration. Therefore, the parameters of the two motors are obtained based on the driving condition analysis for the real-road driving data and multi driving cycles.

### 3.2.1. Statistical Analysis of Typical Driving Conditions

Based on the dynamic characteristics of the three working modes, the MG1 and MG2 are suitable for the urban driving conditions and suburban driving conditions correspondingly. Therefore, 3 urban and 3 suburban driving cycles are selected for the driving condition analysis. At the same time, considering the statistical accuracy, complex fuel consumption test driving conditions such as FTP72 (UDDS) and WLTC cycles are added for analysis. The statistics of selected typical driving conditions are shown in Table 3.

**Table 3.** Data statistics of typical vehicle driving conditions.

| Driving Condition | Cycle Name | Average Speed, km/h | Speed Range/Proportion | Maximum Power, kw | Power Range/Proportion |
|---|---|---|---|---|---|
| Urban driving | China_CITY | 16.1 | 0~30 km/h/(80.29%) | 15.9 | 0~15 kw/(75.26%) |
| | WVUCITY | 13.6 | 0~30 km/h/(79.63%) | 20.9 | 0~15 kw/(77.34%) |
| | ECE | 18.4 | 0~35 km/h/(76.41%) | 17.3 | 0~15 kw/(80%) |
| Suburban driving | EUDC | 62.6 | 50~80 km/h/(56.25%) | 41.2 | 0~25 kw/(84.5%) |
| | WVUINTER | 54.8 | 40~90 km/h/(43.85%) | 33.3 | 0~25 kw/(83.51%) |
| | INDIA_HWY_SAMPLE | 47.6 | 40~70 km/h/(67.92%) | 30.7 | 0~20 kw/(68.03%) |
| Combined driving | UDDS | 31.5 | 0~35 km/h/(50.66%) | 39.4 | 0~15 kw/(64.16%) |
| | WLTC | 46.5 | 30~80 km/h/(41.42%) | 50.2 | 0~30 kw/(69.18%) |

### 3.2.2. Statistical Analysis of Real-Road Driving Conditions

Since the dual motor configuration is designed for the Chinese city driving, the real-road driving experiments are carried out to obtain the urban and suburban driving conditions of Zibo City in China. The real-road driving condition analysis is also applied to the dual motor parameter matching.

Based on the handheld GPS (Magellan eXplorist 510 San Dimas, CA, USA) and the commercial map software, this study obtains the real-road driving data for urban and suburban driving conditions. The collected urban driving conditions of Zibo City (Zibo_urban) and suburban driving conditions of Zibo City (Zibo_suburban) are shown in Figure 2. The statistical analysis is shown in Table 4.

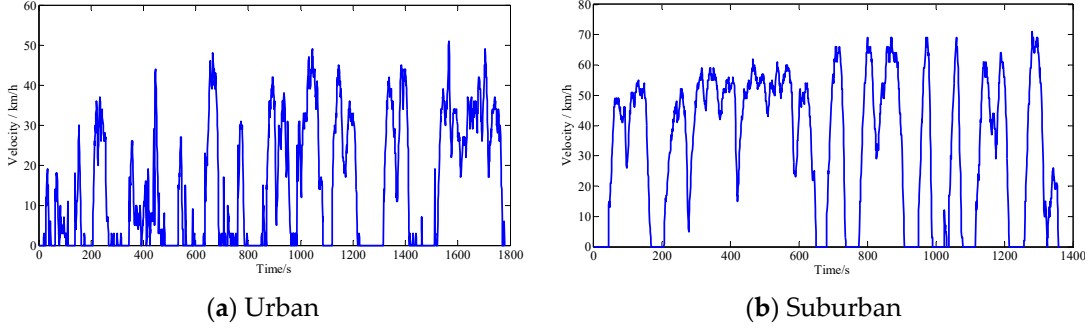

(**a**) Urban　　　　　　　　　　　　　　　　(**b**) Suburban

**Figure 2.** Driving conditions of Zibo City.

**Table 4.** Statistical analysis of real-road driving conditions.

| Driving Cycle Condition | Average Speed, km/h | Speed Range/Proportion | Maximum Power, kw | Power Range/Proportion |
|---|---|---|---|---|
| Zibo_urban | 14.5 | 0~30 km/h/(75.3%) | 31.9 | 0~15 kw/(65.81%) |
| Zibo_suburban | 33.5 | 40~65 km/h/(49.63%) | 55.5 | 0~25 kw/(64.66%) |

### 3.2.3. Real-Road/Driving Cycle Analysis Based Dual Motor Parameter Matching

In general, when the motor operates around the working points of rated power and rated speed, the motor efficiency is high. Therefore, the motor mostly operated high efficiency range should be consistent with the corresponding driving conditions. The motor rated power should satisfy the major power requirement of the corresponding driving conditions.

(1)　Optimal MG1 parameter matching

　　The MG1 parameters are designed based on the dynamic characteristics and suitable driving conditions of MG1 driving mode. The operation of MG1 should be determined by the urban driving conditions of typical driving cycles and real road driving data (China_CITY, WVUCITY, ECE, and Zibo_urban). For the above driving conditions, the MG1 is mostly operated at low speed range. According to the analysis at Tables 3 and 4, it is concluded that the vehicle is mostly driving under the velocity of 30 km/h for around 80% time of the urban driving conditions. In addition, the vehicle demand power is almost under 15 kw around 80% and 66% time of the typical driving cycles and real-road driving data respectively. Besides, the high frequency power range of UDDS integrated test condition is still between 0~15 kw.

　　Since the high frequency demand power under the urban driving conditions should be consistent with the rated power of the MG1, the MG1 rated power can be determined as 15 kw. The maximum demand power for the urban driving conditions is mostly within 30 kw and the calculated maximum climbing power at low speed is under 30 kw. Therefore, the MG1 peak power is determined as 30 kw.

　　Based on the analysis of the urban driving conditions, the high frequency speed range is applied into the Equation (13) to obtain the rated speed of MG1. For the MG1 driving mode, the MG1 is designed to satisfy the vehicle starting and climbing requirement, so the MG1 maximum torque should satisfy the dynamic requirements of the maximum climbing degree.

$$u = 0.377 \cdot \frac{rn}{i_{MG\_X}} \tag{13}$$

$$T_{MG1\_max} \geq \frac{\left(mgfcos\alpha_{max} + mgsin\alpha_{max} + \frac{C_D A u_i^2}{21.15}\right)r}{i_{MG\_X}\eta_t} \tag{14}$$

where $i_{MG\_X}$ is the total transmission ratio when the MG1 working.

(2)　Optimal MG2 parameter matching

　　Compared with the MG1 driving mode, the torque of MG2 driving is lower and the speed of MG2 is higher. Consequently, the MG2 driving mode is more suitable for medium speed operation, and the MG2 parameters need to be determined by suburban driving conditions of typical driving cycles and real road driving data (EUDC, WVUINTER, INDIA_HWY_SAMPLE and Zibo_suburban). From the above statistics, it is known that the MG2 driving mode is suitable for medium speed operation and the high frequency speed range of WLTC driving cycle should be considered. Consequently, the high proportion middle speed range between 40~80 km/h is selected as the reference although the proportion of low speed range is still very high.

　　According to Tables 3 and 4, it can be found that the demand power is almost under 30 kw and the maximum demand power for the above suburban driving conditions is 55.5 kw. Accordingly, the MG2 peak power is determined as 60 kw and the MG2 rated power is set as 30 kw, which can better meet the power demand for the medium speed operation. The high-frequency speed range of 40~50 km/h in the above suburban conditions is applied to the Equation (13) to determine the MG2 rated speed. Where $i_{MG\_X}$ is the total transmission ratio when the MG2 working.

*3.3. Parameter Matching and Optimization of Transmission System*

3.3.1. Preliminary Parameter Matching of Transmission System

For the dual motor coupling configuration, the maximum transmission ratio of the target vehicle depends on the maximum gradient, and the minimum transmission ratio depends on the maximum speed. The equation is shown as:

$$\frac{mg(f cos\alpha_{max} + sin\alpha_{max})r}{T_{max}\eta_t} \leq \sum i \leq \frac{0.377 n_{max}r}{u_{max}} \tag{15}$$

Due to the influence of the mechanical structure of the planetary gear, its characteristic parameter is usually taken as $1.5 \leq k \leq 4$. So the characteristic parameter of the planetary gear $k$ is selected as 2.6, $i_0$ is 3.7, and $i_1$ is 1.9.

3.3.2. Transmission Parameters Optimization Based on Particle Swarm Optimization

Among all the optimization algorithms, particle swarm optimization (PSO) has the advantages of easy implementation, high precision and fast convergence speed [20], so PSO is used to optimize the parameters of the transmission system globally.

(1)  Optimization variables

The transmission system has the function of adjusting speed and increasing torque. Reasonable design of transmission system parameters can effectively improve the dynamic performance and the economy of the target vehicle. Therefore, in this paper, only the characteristic parameters of planetary gears $k$, fixed gear ratio $i_1$ and final reduction gear ratio $i_0$ are taken as optimization variables:

$$X = [X\,(1), X\,(2), X\,(3)]^T = [k, i_1, i_0]^T \tag{16}$$

(2)  Objective function

In order to obtain the maximum system efficiency, the minimum energy consumption under multiple driving conditions (e.g., China_CITY driving condition, Zibo_suburban and UDDS driving condition) is selected as the objective function.

$$Q = \frac{Pt}{\eta_{motor}} = \int_0^t \frac{1}{\eta(t)_{motor}} \cdot \Delta P dt \tag{17}$$

$$\Delta P = \frac{u(t)}{3600\eta_t}\left(mgf + \frac{C_D \times A \times u(t)^2}{21.15} + \delta m \frac{du}{3.6 \times dt}\right) \tag{18}$$

where $P$ is the motor power; $u(t)$ is the corresponding speed at time t; $\eta(t)$ is the system efficiency at time $t$, which is determined by the MG1, MG2 and transmission efficiency at time $t$.

(3)  Constraints

The maximum climbing degree, the maximum speed, and the maximum adhesion and component characteristic parameters are taken as constraints to ensure the dynamic performance. For the MG1 driving mode, the MG1 should satisfy the requirements of maximum climbing gradient. At the same time, the corresponding driving force should be less than the requirements of maximum adhesion. For the dual-motor speed coupling mode, the maximum vehicle velocity should be considered.

$$\frac{mg(f cos\alpha_{max} + sin\alpha_{max})r}{T_{max}\eta_t} \leq i_0(k+1) \tag{19}$$

$$i_0(k+1) \le \frac{F_z \phi r}{T_{max} \eta_t} \tag{20}$$

$$i_0 \le \frac{0.377 \times \left( \frac{n_{MG1\_max}}{1+k} + \frac{k \times n_{MG2\_max}}{i_1(1+k)} \right) \times r}{u_{max}} \tag{21}$$

where $T_{max}$ is the maximum torque output when the MG1 working; $F_Z$ is the normal reaction force of the ground to the tire, and the front and rear axle loads are average distributed, which is 0.5; $\phi$ is the road adhesion coefficient, which is 0.75; $n_{MG\_max}$ is the maximum speed of the motor.

The population size is set as 50 and the number of iterations is set as 100. The optimized parameters are shown in Table 5. It can be seen from Table 5 that the energy consumption of the NEDC driving condition are significantly reduced by 7.1%, and the maximum climbing degree is also improved by 5.7%. In order to guarantee the vehicle dynamic performance, the maximum speed of the system is relatively reduced. On the basis of ensuring dynamic performance, the transmission parameters with less energy cost are selected.

**Table 5.** Parameter comparison after optimization.

| Item | Primary Parameters | Optimized Parameters | Improvement |
|---|---|---|---|
| Characteristic parameter, $k$ | 2.6 | 2.4 | - |
| Fixed gear ratio, $i_1$ | 1.9 | 1.8 | - |
| Final reduction gear ratio, $i_0$ | 3.7 | 4.1 | - |
| Maximum speed, km/h | 144 | 136 | −5.6% |
| Maximum climbing grade | 35 | 37 | 5.7% |
| Energy consumption under NEDC condition, kwh | 2.26 | 2.10 | 7.1% |

In conclusion, the dynamic parameters of the whole vehicle calculated according to the above theory are shown in Table 6.

**Table 6.** Vehicle parameters.

| Item | Value | Unit |
|---|---|---|
| Rated/peak power (MG1) | 15/30 | kw |
| Rated/maximum torque | 70/145 | Nm |
| Rated/maximum speed | 2000/6200 | rpm |
| Rated/peak power (MG2) | 30/60 | kw |
| Rated/maximum torque | 95/190 | Nm |
| Rated/maximum speed | 3000/7500 | rpm |
| Planetary gear characteristic parameters, $k$ | 2.4 | - |
| Final reduction gear ratio, $i_0$ | 4.1 | - |
| Fixed gear ratio, $i_1$ | 1.8 | - |
| Battery capacity | 145 | A·h |
| Rated voltage of battery | 360 | V |

## 4. Development of Full Vehicle Model Using MATLAB/Simulink

In this study, the full vehicle model of the target dual motor configuration is developed using MATLAB/Simulink. The full vehicle model consists of the component dynamic model and controller model.

*4.1. Development of Vehicle Powertrain Model*

The motor, battery, planetary gear, reduction gear, vehicle and driver models are developed using MATLAB SimDriveline module. The SimDriveline based powertrain model is already verified using experiment data in the previous study [21]. Therefore, this paper only displays the development of motor and battery models. In this study, the powertrain model is validated to verify the system performance of the target vehicle.

### 4.1.1. Development of Motor Model

The motor model is constructed according to the motor characteristic map and motor efficiency map. The corresponding equations are as follows:

$$T_{motor} = AP \times f(\omega_{motor}) \tag{22}$$

$$\eta_{motor} = f(\omega_{motor}, T_{motor}) \tag{23}$$

where $AP$ is acceleration signal; $f(\omega_{motor})$ is implies as the motor characteristic map with the input of motor angular speed; $\eta_{motor}$ is the motor efficiency, $f(\omega_{motor}, T_{motor})$ is implies as the motor efficiency map with the input of motor angular speed and motor torque. The efficiency maps of MG1 and MG2 are shown in Figure 3.

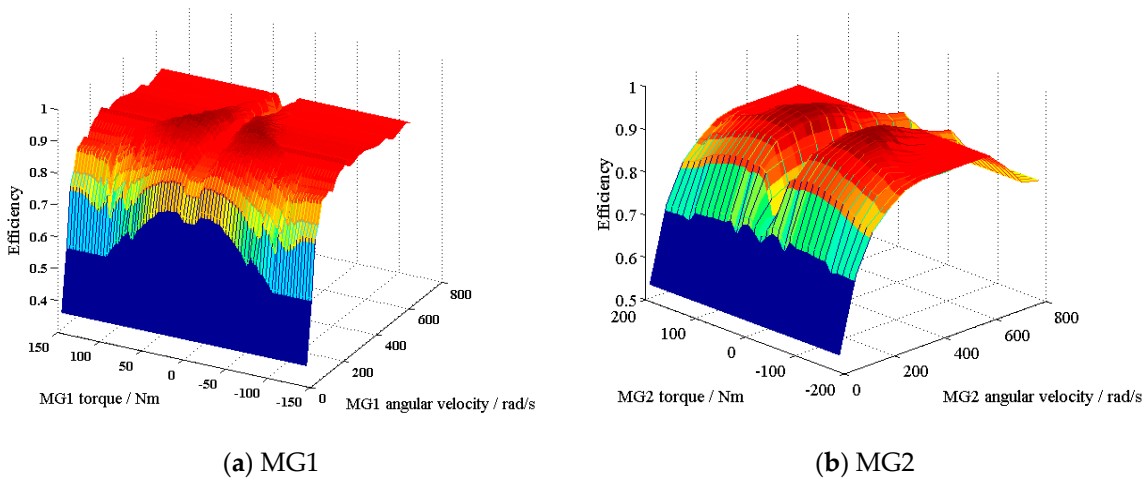

(**a**) MG1                    (**b**) MG2

**Figure 3.** Efficiency map of motor.

### 4.1.2. Development of Battery Model

The battery model is developed by the current integration method:

$$SOC = \frac{C - C_{use}}{C} \tag{24}$$

$$C_{use} = \int Idt + (1 - SOC_{init}) \times C \tag{25}$$

where $C_{use}$ is the battery capacity used; $I$ is the current, and $SOC_{init}$ is the initial state of charge. In addition, the battery voltage and internal resistance are obtained using battery voltage map and battery internal resistance map.

### 4.2. Development of Mode Shift Algorithm

(1)  Mode shift control

The mode shift algorithm is developed considering the dynamic characteristics of the three working modes and the analysis on the real-road/typical driving cycles. Based on the analysis above, it can be concluded that the MG1/MG2/dual-motor coupling driving modes correspond to low speed, medium speed and high speed respectively. Hence, a rule-based energy management strategy can be developed. When the MG1 works under 30 km/h, it can meet around 80% time of the low-speed urban driving conditions. Therefore, when the speed is less than 30 km/h and the demand power is less than the MG1 peak power, the MG1 driving alone. Similarly, other rules can also be inferred as shown in the Figure 4.

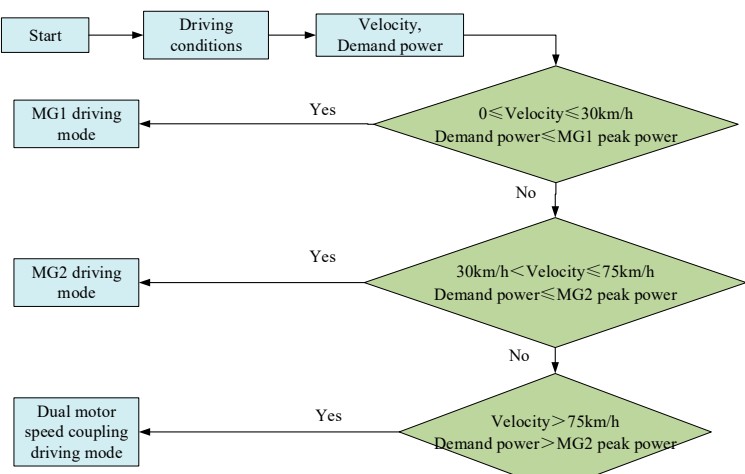

**Figure 4.** Mode shift strategy.

(2)　Optimal dual motor speed operating control

For the dual motor coupling driving mode, the MG1 and MG2 are cooperated together to propel the vehicle. According to dynamic characteristics analysis at Equation (5), it is known that even if the vehicle is driving under some certain velocity, the MG1 and MG2 can cooperate together to propel the vehicle with various combinations of the two motor speeds. Consequently, the MG1 and MG2 efficiency can be different for different MG1 and MG2 speed combinations, which causes the different system efficiency at same vehicle speed.

In order to improve the system efficiency of dual motor coupling mode, an optimal dual motor speed operating control algorithm is proposed. The optimal speed of MG1 and MG2 under different speed and torque requirements is determined based on the optimal system efficiency of the whole vehicle, and the maximum vehicle speed, maximum speed and maximum torque are constraints:

$$
\begin{aligned}
&Obj : Max\{\eta_{dual\_drive}\left(V, T_{driveshaft}, \omega_{MG1}, \omega_{MG2}\right)\}\\
&Subject\ to:\\
&0 \le V \le V_{max}\\
&0 \le T_{driveshaft} \le T_{max}\\
&-T_{MG1\_max} \le T_{MG1} \le T_{MG1\_max}\\
&-T_{MG2\_max} \le T_{MG2} \le T_{MG2\_max}\\
&-\omega_{MG1\_max} \le \omega_{MG1} \le \omega_{MG1\_max}\\
&-\omega_{MG2\_max} \le \omega_{MG2} \le \omega_{MG2\_max}
\end{aligned}
\tag{26}
$$

where $\eta_{dual\_drive}$ is dual motor driving efficiency; $T_{driveshaft}$ is driving shaft torque; $V_{max}$ is maximum vehicle speed; $T_{MG\_max}$ is the maximum torque of the motor; $\omega_{MG\_max}$ is the maximum speed of the motor; $T_{max}$ is the maximum torque of the system.

The process of obtaining the optimal speed of dual motor is shown in Figure 5.

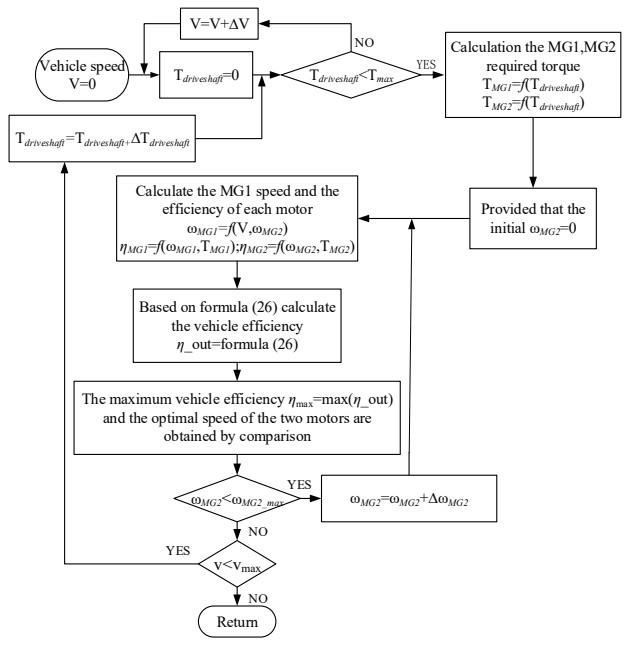

**Figure 5.** Flow chart for obtaining optimal speed of dual motor.

## 5. Simulation Analysis

In order to verify the correctness of the parameter matching results, the data of the model is defined according to the above parameter matching results. Then, the UDDS cycle is selected as the test working condition to verify the correctness of the simulation model. Finally, the improvement of the proposed configuration compared with the single motor configuration is compared.

### 5.1. Simulation Results of UDDS Cycle

The UDDS cycle is selected for the full vehicle performance analysis. The velocity following status and motor working status are shown in Figure 6.

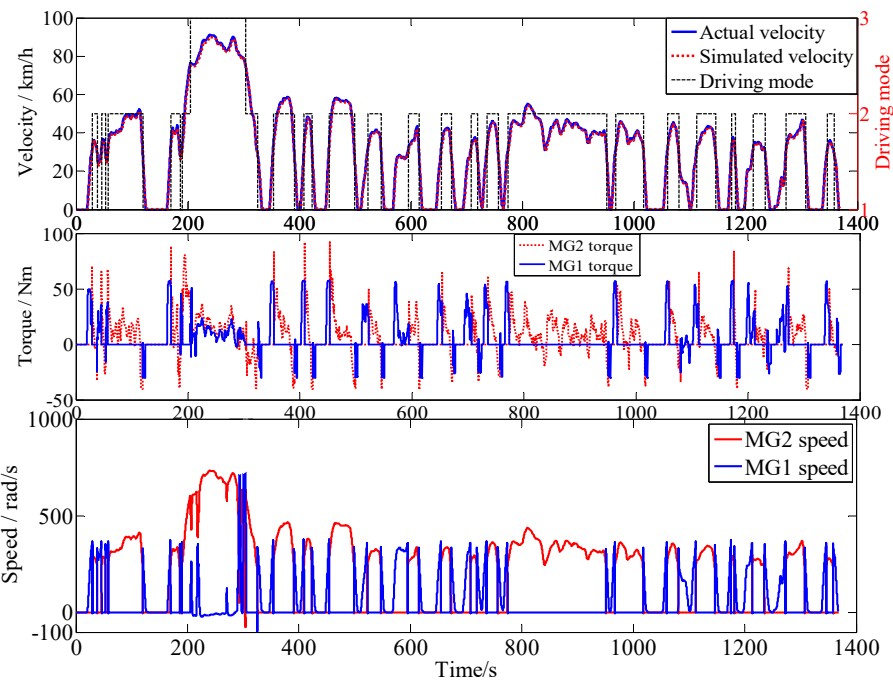

**Figure 6.** Velocity and motor working status chart.

The simulation results show that the velocity can follow the actual velocity closely and the vehicle is mostly driving under MG2 driving mode, which is also consistent with the analysis on the UDDS cycle that the speed of UDDS is mostly between 30~75 km/h. When the vehicle speed reaches 75 km/h at time 203 s, the driving mode is switched from MG2 driving mode to dual-motor speed coupling mode. Moreover, at *t* = 203–303 s, it can be seen from the motor speed/torque that the MG1 and MG2 cooperate together to propel the vehicle. Owing to the application of dual motors coordinate control, the dual motor achieves the optimal working at optimal speed. It avoids the low efficiency problem when the single motor driving speed is too high and realizes the improvement of the system efficiency. From the motor torque curve, it is seen that the MG1 output torque to propel the vehicle for the velocity of 0~30 km/h, MG2 output torque to propel the vehicle for the velocity of 30~75 km/h and the MG1 and MG2 cooperate together to propel the vehicle when the velocity is higher than 75 km/h with the time of 203~303 s.

From the above analysis, it can be concluded both the parameter matching and the proposed mode shift strategy are correct and validated by the simulation results.

### 5.2. Comparative Analysis between Single Motor and Dual Motor Configuration

In order to verify the efficiency improvement of the dual motor coupling configuration, a comparative analysis is performed between the single motor and dual motor coupling configuration. The motor working points, battery SOC (State of Charge), and energy consumption simulation results are compared and analyzed respectively.

(1) Comparative analysis of motor working points

The motor working points for the two configurations are shown in Figures 7 and 8.

For the single motor configuration, it is seen from Figure 7 that the motor operating points are mostly in the low efficiency area, and the torque load of the motor is also low, which causes additional energy losses. For the dual-motor coupling configuration, it is seen from Figure 8 that motor working points move to the high efficiency area and torque load of the motor is increased, which implies the correctness of the real-road and typical driving-cycle analysis based parameters matching of the motor and transmission system.

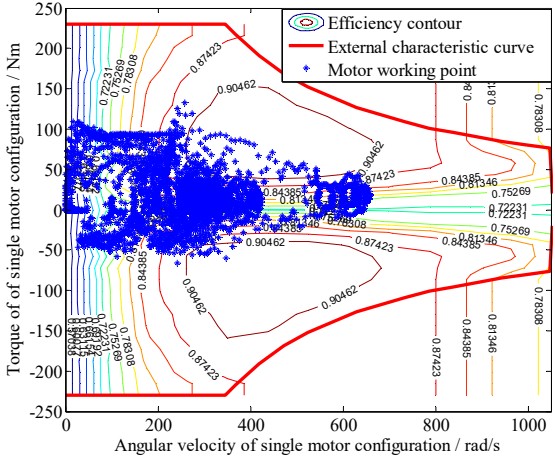

**Figure 7.** Motor working points of single motor configuration.

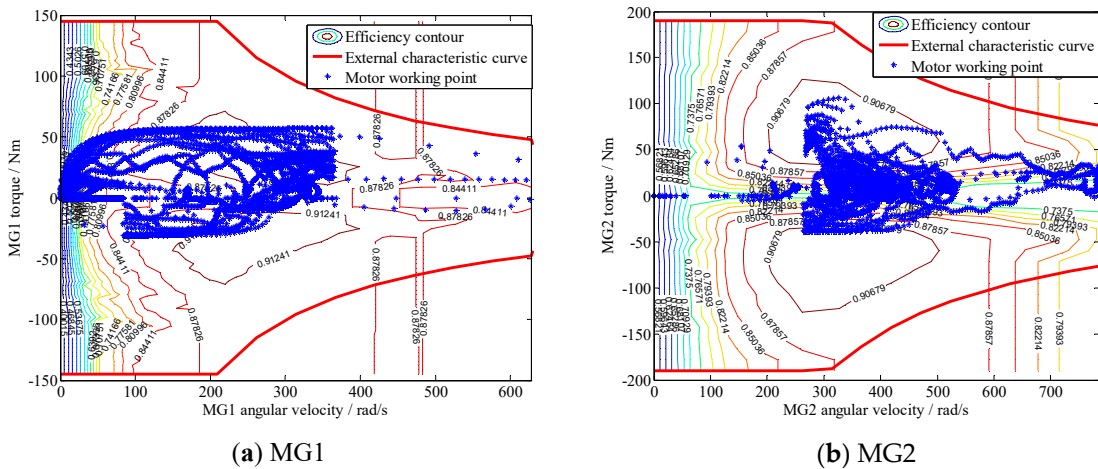

(**a**) MG1      (**b**) MG2

**Figure 8.** Motor working points of dual motor coupling configuration.

(2)   Comparative analysis of final battery SOC

The battery SOC for the two configurations is shown in Figure 9 with the initial SOC of 0.9 for 5 UDDS cycles.

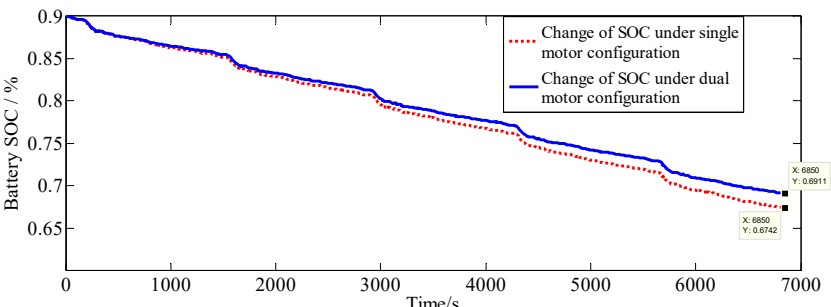

**Figure 9.** Battery SOC curve under two configurations.

As shown in Figure 9, the battery SOC has a gradual downward trend under two configurations. The final battery SOC of the single motor and dual motor coupling configuration is 0.6742 and 0.6911 respectively, which indicates energy consumption reduction of 7.6% for the dual motor coupling configuration.

(3)   Comparative analysis of energy consumption for various test conditions

In order to further verify the effectiveness under other driving conditions, NEDC, WLTC, Zibo_urban and Zibo_suburban are selected for simulation analysis. The simulations of the above driving conditions are all running under the full load condition of the vehicle. Since slope of collected Zibo driving condition is relatively small, the slope can be approximately zero, so the effect of slope on vehicle energy management can be ignored. The initial battery SOC is set as 0.9 and the simulation results are listed in Table 7.

The simulation results show that under the low-speed congestion condition of Zibo City, the energy-saving effect of the dual motor configuration is significant. Since the proposed configuration has a small size motor, the dual motor configuration can avoid the low efficiency problem for low speed operation and achieve energy saving driving for low speed driving conditions. Compared with the single motor configuration, the energy consumption of the proposed configuration is reduced by 16.3% for Zibo City driving conditions.

**Table 7.** Energy saving effect under other driving conditions.

| Driving Conditions | NEDC | UDDS | WLTC | Zibo_Urban and Zibo_Suburban |
|---|---|---|---|---|
| Final SOC (single motor configuration) | 0.8576 | 0.8548 | 0.7860 | 0.8152 |
| Energy consumption/kwh | 2.21 | 2.36 | 5.95 | 4.42 |
| Final SOC (dual motor configuration) | 0.8597 | 0.8582 | 0.7930 | 0.8291 |
| Energy consumption/kwh | 2.10 | 2.18 | 5.58 | 3.70 |
| Cost reduction | 5.0% | 7.6% | 6.2% | 16.3% |

In addition, under medium speed driving conditions, the energy consumption of the proposed configuration is reduced by 5.0% and 7.6% respectively for NEDC and UDDS cycle. Furthermore, the coordinated control of the dual motor under high speed driving condition also achieves the optimal operation of two motors with high system efficiency. Therefore, the proposed scheme also has a significant improvement under higher speed driving conditions. The energy consumption of the proposed configuration is reduced by 6.2% for WLTC driving cycle.

From the analysis above, it is found that the proposed dual motor coupling configuration achieves better energy saving for various driving conditions.

## 6. Conclusions

In this paper, a planetary gear based dual motor coupling configuration EV is proposed, which can realize efficient multi modes operation for various driving conditions.

According to the dynamic analysis and the statistics analysis on typical driving cycles/real-road driving data, the motor optimal parameters are obtained, which reduces the motor power redundancy and improves the motor working efficiency. Then, a particle swarm optimization algorithm based transmission system parameter optimization is performed with the goal of minimum energy consumption which achieves a reduction in energy consumption of 7.1% compared to the parameters before optimization. Besides, the establishment of mode shift algorithm and the optimal dual motor speed operating control further improves the EV economy.

The simulation results indicate that the motor working points are closer to the high efficiency area and the motor load rate is higher under the dual motor configuration. The battery SOC drops slowly under different driving conditions, which achieves the economy improvement of 5~16.3%. Especially under the low speed driving condition, the energy saving effect is remarkable, and the driving range is significantly extended. From the above conclusions, it can be confirmed both the proposed configuration and the parameter matching are correct and valid, and the EV economy can be comprehensively improved.

**Author Contributions:** Conceptualization, C.M.; methodology, C.M. and S.J.; software, K.Y.; validation, D.T.; investigation, J.G. and D.Y.; writing—original draft preparation, S.J.; writing—review and editing, C.M. All authors have read and agreed to the published version of the manuscript.

**Funding:** This work was supported by the National Natural Science Foundation of China (Grant No. 51605265 and 51775320), and the Key Research and Development Program of Shandong Province of China (Grant No. 2019GGX104069).

**Conflicts of Interest:** The authors declare no conflict of interest.

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
