# Peer review of "Particle Swarm Optimization and Real-Road/Driving-Cycle Analysis Based Powertrain System Design for Dual Motor Coupling Electric Vehicle"

_wevj, doi:10.3390/wevj11040069_

Round 1

Reviewer 1 Report

The introduction is good, however, it can be improved. The authors must present in the introduction very clearly the purpose and objectives of the paper.

Motors working modes (MG1 & MG2): the authors must present the algorithm of operation of the two electric motors (independently or simultaneously) depending on the speed of the vehicle, acceleration, loading (load) of the vehicle, altitude profile of the route, environmental climatic conditions. Table 2 explains what happens when the motors run independently or simultaneously, but we do not know under what conditions this happens. The details presented in line 99 to 118 (The dynamic characteristics of each working mode are derived as follows...) are not very clear.

Data statistics of typical vehicle driving conditions (considered driving cycles) do not contain statistics, about two of the most current, important and complex travel cycles WLTC (Worldwide Harmonized Light Vehicles Test Cycle) and FTP (EPA Federal Test Procedure) version for electrical vehicles. In order to increase the accuracy of the results, I consider that it was very important if these travel cycles were implemented in the evaluation of the performance of the electric vehicle.

The working methodology is correct, the real data collected in urban and sub-urban traffic were used to define the virtual vehicle model in MATLAB-Simulink. The approach for the simulation process methodology is also good: Motor working points of single motor configuration vs. Motor working points of dual motor coupling configuration. But, it would have been useful to see an evaluation of the energy efficiency for the consumption of electricity in the operation of the customized electric vehicle for all the real operating conditions presented above (speed of the vehicle, acceleration, the load of the vehicle, altitude profile of the route, environmental climatic conditions, traffic, rush hours etc.). The results presented are very few. Discussions about these are almost non-existent. I recommend the authors to return to them.

The conclusions of the paper are very brief. I recommend the authors to complete these conclusions.

Author Response

Dear Editors and Reviewer,

Thank you very much for taking your time to review this manuscript. I really appreciate all your comments and suggestions! Please find my itemized responses in below and my revisions/corrections in the re-submitted files. The corresponding modifications are marked with yellow color.

Point 1: The introduction is good, however, it can be improved. The authors must present in the introduction very clearly the purpose and objectives of the paper.

Response 1: For your suggestion, we add some sentences to make the introduction more compact, and highlight the goal and purpose of the article. (Line 43 to 44, line 49 to 50, line 65 to 67, line 69 to 70 and line 75 to 79)

Point 2: Motors working modes (MG1 & MG2): the authors must present the algorithm of operation of the two electric motors (independently or simultaneously) depending on the speed of the vehicle, acceleration, loading (load) of the vehicle, altitude profile of the route, environmental climatic conditions. Table 2 explains what happens when the motors run independently or simultaneously, but we do not know under what conditions this happens. The details presented in line 99 to 118 (The dynamic characteristics of each working mode are derived as follows...) are not very clear.

Response 2: Thank you for the suggestion. For the shift control of the working mode, it is obtained by the statistical analysis of the later driving condition data and the parameter matching results. The detailed explanations are shown in Chapter 4.2 (Development of mode shift algorithm, page 11).

In order to make the explanations of the dynamic characteristics for each working mode more clearly, the energy transfer of each working mode is added and explained (line 106, line 108 to 111, line 114 to 117 and line 120 to 122). The symbols of the formula are further explained, so that the formula can be better understood (Line 125 to 127).

Point 3: Data statistics of typical vehicle driving conditions (considered driving cycles) do not contain statistics, about two of the most current, important and complex travel cycles WLTC (Worldwide Harmonized Light Vehicles Test Cycle) and FTP (EPA Federal Test Procedure) version for electrical vehicles. In order to increase the accuracy of the results, I consider that it was very important if these travel cycles were implemented in the evaluation of the performance of the electric vehicle.

Response 3: Thank you for the suggestion, we add statistical analysis of FTP72 (UDDS) and WLTC driving cycles (in table 3, page6), and make some analysis on them (Line183 to 185, line 214 to 215, line 234 to 235 and line237).

Similarly, the comparison of simulation results of UDDS driving mode is added in the simulation results below (in table 7, page15), which makes the results more convincing.

Point 4: The working methodology is correct, the real data collected in urban and sub-urban traffic were used to define the virtual vehicle model in MATLAB-Simulink. The approach for the simulation process methodology is also good: Motor working points of single motor configuration vs. Motor working points of dual motor coupling configuration. But, it would have been useful to see an evaluation of the energy efficiency for the consumption of electricity in the operation of the customized electric vehicle for all the real operating conditions presented above (speed of the vehicle, acceleration, the load of the vehicle, altitude profile of the route, environmental climatic conditions, traffic, rush hours etc.). The results presented are very few. Discussions about these are almost non-existent. I recommend the authors to return to them.

Response 4: Thank you for your suggestion. The definitions of vehicle load condition, as well as the altitude and gradient of the area are added in line 409 to 412 (Page 14). Simultaneously, and the simulation results are discussed separately according to the vehicle speed, which makes the results more clear. The improvement of the vehicle under different speed driving conditions becomes more obvious. The modification result is shown in line 418 to 423 (page 15). The impact of environment, climate and slope, etc. on energy management strategy will be studied in the future.

Point 5: The conclusions of the paper are very brief. I recommend the authors to complete these conclusions.

Response 5: Thank you for the suggestion. We further expand the results to make the results more accurate and rich. The modification results are shown in line 435, line 439 to 441 and line 445 to 448.

Reviewer 2 Report

English check is required.

The markings in the efficiency maps may be improved to be clearer.

Some figures, e.g. Figure 6, are not clear, especially hindered by the heavy motor working points.

In the section “Optimal dual-motor speed operating control”, it may be better to include optimal result to show the best combination and how the optimal result is actually obtained.

Author Response

Dear Editors and Reviewer,

Thank you very much for taking your time to review this manuscript. I really appreciate all your comments and suggestions! Please find my itemized responses in below and my revisions/corrections in the re-submitted files. The corresponding modifications are marked with yellow color.

Thank you and best regards!

Point 1: English check is required.

Response 1: Thank you for the suggestion. We checked the English in the article and made some revisions.

Point 2: The markings in the efficiency maps may be improved to be clearer.

Response 2: Thank you for underlining this deficiency. We modify the motor efficiency map to make it clearer, e.g. Figure 3, Figure 7 and Figure 8. Similarly, we changed all the figures of the article to a clearer version (figure 1 to figure 9).

Point 3: Some figures, e.g. Figure 6, are not clear, especially hindered by the heavy motor working points.

Response 3: In response to this proposal, for the pictures mentioned, we reduced the motor working point density and motor efficiency contour density to make the figure more visible (Figure 6 is changed to Figure 7).

Point 4: In the section “Optimal dual-motor speed operating control”, it may be better to include optimal result to show the best combination and how the optimal result is actually obtained.

Response 4: Thank you for the suggestion. We add the flow chart of obtaining the optimal speed of dual motor, so that this part can be better understood. (figure 5, page 12)

In the following simulation results (figure 6, page 13), the dual motor optimal simulation results are added and shown. During t=203-303s, it can be seen that the MG1 and MG2 speeds are non-zero at the same time, which implies that the MG1 and MG2 are cooperated together to propel the vehicle. The explanations of optimal control results of the dual motor are added in line 368 to 372.

Round 2

Reviewer 1 Report

The authors responded to the requests in the review and modified the paper for the better. In conclusion, after a review of the writing style and the English language, I recommend the publication of the paper.